# Identification of the Transcription Factors RAP2-13 Activating the Expression of *CsBAK1* in Citrus Defence Response to *Xanthomonas citri* subsp. *citri*

Qi Wu [1,2], Mingming Zhao [1,2], Yi Li [1,2], Dazhi Li [1,2], Xianfeng Ma [1,2] and Ziniu Deng [1,2,*]

1 College of Horticulture, Hunan Agricultural University, Changsha 410128, China
2 National Center for Citrus Improvement, Changsha 410128, China
* Correspondence: deng7009@163.com

**Abstract:** Citrus canker is a quarantined disease caused by the bacterial plant pathogen *Xanthomonas citri* subsp. *citri* (*Xcc*), which causes persistent surface damage, leaf and fruit drop, and tree decline in citrus plants. The citrus cultivar Citron C-05 (*Citrus medica* L.) is a disease-resistant genotype identified after years of screening at the National Center for Citrus Improvement (Changsha), which displays allergic, necrotic, and disease-resistant responses to *Xcc*. In this study, the *BAK1* gene was identified in this cultivar to be a disease resistance gene involved in plant-microbe interaction between citrus and *Xcc*. Functional investigations of this gene revealed that both *CsBAK1 (C. sinensis BAK1)* or *CmBAK1(C. medica BAK1)* could inhibit the growth of *Xcc* to some extent when transiently expressed in the susceptible 'Bingtang' genotype of sweet orange. Critical regions of the *CmBAK1* promoter sequence were identified by creating downstream deletions and exposing mutants to *Xcc* to determine effects on the resistance phenotype; a 426 bp region (−2000~−1574) was identified as a key functional region responsible for eliciting the hypersensitive response in plants. Through screening arrayed Citron C-05 cDNA libraries by yeast one-hybrid assays, a basic APETALA2/ETHYLENE RESPONSE FACTOR (AP2/ERF) transcription factor of *CmRAP2-13* that binds directly to the 426 bp key sequence and activates expression of *CmBAK1* was identified. Moreover, transcriptional analysis revealed an obvious increase in transcript levels of *CsRAP2-13* in Citron C-05, American citron, and Finger citron. In this study, we present the identification of transcriptional activators that are found to interact with BAK1 proteins in response to *Xcc*. These results reveal a coordinated regulatory mechanism of RAP2-13, which may be involved in defence responses through the regulation of BAK1.

**Keywords:** *Xcc*; *CsBAK1*; *CsRAP2-13*; disease resistance; transcription factor

## 1. Introduction

Citrus canker is a devastating disease caused by the Gram-negative bacterium *Xanthomonas citri* subsp. *citri* (*Xcc*). The main symptoms of citrus canker are the formation of crater-like disease spots on the surface of branches, leaves, and fruits, which can lead to leaf and fruit drop, retardation of tree growth and development, decreased yield, and cosmetic damage that makes fruit unmarketable [1,2]. At present, the control of citrus canker relies primarily on chemical applications; however, disease eradication using this method is challenging and potentially environmentally damaging. Breeding resistant citrus varieties is widely considered a more effective and safe control measure [3]. Investigating the interactions between citrus plants and *Xcc* to identify molecular components underpinning the resistance of citrus to this pathogen greatly helps inform breeding program efforts aimed at developing disease resistant cultivars.

Plants have a variety of strategies to defend against invading plant pathogens, including increasing the strength of their cell walls, producing specific metabolites to kill pathogens, and regulating defence-related gene expression in plant cells. It is now widely recognized that plants possess a system of innate immunity. When plants are exposed to

potential pathogens, they are able to resist many diseases. Plants can become susceptible lose such disease resistance when a pathogen produces a specific effector that can bypass the plant defence system [4]. Extracellular recognition of microbial and host damage associated molecular patterns (MAMPs and DAMPs, respectively) by plants induces the first layer defence known as pattern-triggered immunity (PTI).

Citrus has several reproductive characteristics, such as long juvenility, self-incompatibility, nucellar embryony, heterozygosity, and apomixis, which hinder efforts to develop resistant varieties through conventional breeding. Genetic modification of plants involves the intentional alteration of the plant genome through the insertion of foreign DNA sequences [5]. Resistance genes from the same or different species can be transgenically integrated into susceptible citrus cultivars and have the potential to improve the PTI response and resistance to citrus canker of these susceptible species [6]. FLS2 is a good candidate resistance gene for integration into citrus improvement programs. For example, overexpression of Xa21 in the sweet orange cultivar 'Dark' resulted in increased tolerance to *Xcc* [7].

A common coreceptor found in plants involved in the regulation of specific cellular mechanisms such as growth, development, and defence against pathogens is the membrane-bound brassinosteroid-insensitive associated receptor kinase BAK1. BAK1 attaches to ligand stimulated transmembrane receptors, thereby stimulating their kinase domains through transphosphorylation [8]. Additionally, BAK1 plays a common role in several pathways responsible for PTI signal transduction in *Arabidopsis thaliana* [9]. BAK1 is a leucine rich repeat receptor-like kinase (LRR-RLK). The extracellular LRR domain of BAK1 consists of five repeat sequences, and serine-rich and proline-rich domains are located after these LRR domains [10]. BAK1 also has a transmembrane domain, a cytoplasmic kinase domain, and a short C-terminal tail. As a member of the serine protein family, BAK1 has four homologues. Somatic embryogenic receptor kinase 1 (SERK1) is the first member of this family to be identified, which is why BAK1 is also known as SERK3 [10]. In plant innate immunity, BAK1 plays an important role in communication with the receptor *FLS2* [11,12]. Thus, BAK1 plays a crucial role in the control and management of some LRR-RLKs by cooperating with certain LRR-RLKs in multiple stimulus-dependent systems [13,14]. The important role of *BAK1* in citrus response to invasion by *Xcc* remains unclear.

After years of screening, the cultivar Citron C-05 (*Citrus medica* L.) was identified as a resistant germ plasm that developed no obvious disease and produced hypersensitivity reaction after inoculation with ulcerative bacteria [15,16]. In order to utilize Citron C-05 in breeding for resistance to canker disease, it is necessary to identify the resistance-related genes and explore its resistance mechanism to the disease. This study aims to analyze the expression profiles of the gene *BAK1*, and to better understand the resistant mechanism of citron C-05, which may provide a theoretical basis for disease resistance molecular breeding. In this study, the *BAK1* gene and its promoter were cloned and analyzed in both Citron C-05 and 'Bingtang' sweet orange genotypes, and transcription factors regulating *BAK1* gene expression were screened by yeast monoclonal screening, so as to understand whether *BAK1* is involved in the transcriptional regulation of the defence response to canker disease in the Citron C-05 cultivar.

## 2. Materials and Methods

### 2.1. Plant Materials

Both susceptible citrus genotypes ('Bingtang' sweet orange, Pomelo, Nanchuan citron, Wild citron, and Round citron) and resistant ones (American citron, Finger citron, and Citron C-05) were used in the present study. Plants of all the tested genotypes were grafted onto trifoliate orange (*Poncirus trifoliata*) rootstock and grown for two years in a greenhouse maintained at 28 ± 1 °C, 80% relative humidity (RH), and natural photoperiod. Fully expanded young leaves with light green coloration were selected for *Xcc* inoculation.

## 2.2. Bacterial Pathogen Preparation

The *Xcc* strain DL509, previously isolated and stored at −80 °C in the laboratory, was cultured in LB medium (10 g/L peptone; 10 g/L sodium chloride; 5 g/L yeast extract) at 28 °C for 72 h, and single colonies were selected and cultured in 100 mL liquid LB medium at 28 °C with shaking at 200 rpm/min for 24 h. The bacterial solution was centrifuged at $7000 \times g$ for 10 min. The bacterial pellet was rinsed 3 times with sterile water, and then resuspended in sterile water. The $OD_{600}$ of the bacterial suspension was adjusted to 0.75 (about $10^9$ CFU/mL). After a 10-fold gradient dilution, the $10^{-7}$, $10^{-8}$ and $10^{-9}$ dilution gradients were selected for plate counting, and the $10^{-4}$ ($2.01 \times 10^5$ CFU/mL) dilution gradient was selected for inoculum. Control plants were simultaneously inoculated with the same concentration of *Xoo (Xanthomonasoryzae* pv. oryzae). *Xcc* was injected on one side of the main vein of the sampled fully expanded young leaves and *Xoo* on the other side of the same leaf as a control. Inoculated leaves were sampled at 0, 2, 4, 6, 8, and 10 days post inoculation (dpi) for successive RT-PCR.

## 2.3. RNA Isolation and Quantitative RT-PCR

Total RNA was isolated from sampled leaves using the RNA prep pure plant kit (Tiangen) based on the manufacturer's instructions. cDNA was synthesized from 1 µg of RNA using the PrimeScript RT reagent Kit (Takara, Dalian, China). Quantitative PCR was performed on a Bio-Rad CFX 96 quantitative PCR instrument using SYBR Green I SuperMix (Bio-Rad, Hercules, CA, USA). The reaction procedure was as follows: pre-denaturation at 94 °C for 5 min followed by 40 cycles of denaturation at 94 °C for 5 s, annealing at 59 °C for 15 s, and extension at 72 °C for 20 s. *EF1-α* (Table 1) was used as the reference gene [17], and the $2^{-\triangle\triangle Ct}$ algorithm was used for data calculation.

**Table 1.** *BAK1* and *EF1-α* gene primer sequences for qPCR.

| Gene | Forward Primer (5′-3′) | Reverse Primer (5′-3′) |
|------|------------------------|------------------------|
| *BAK1* | ACCAGAATACGGGCAGACCT | AAGGTTCCCGTCAACTCGTTA |
| *EF1-α* | GTAACCAAGTCTGCTGCCAAG | GACCCAAACACCCAACACATT |

## 2.4. Isolation and Bioinformatic Analysis of the CsBAK1 Gene in Citrus

*BAK1* was ampyfied from cDNA by homologous recombination using specific primers (Table 1). The target fragment was cloned into the pCAMBIA1301 vector (named 35S:: BAK1) and then transformed into *E.coli* strain DH5α (TransGen Biotechnology Co., Ltd., Beijing, China) using heat shock. Positive clones identified by colony PCR were confirmed with Sanger Sequencing (Springen Biotechnology Co., Ltd., Nanjing, China). To isolate *CsBAK1* from genomic DNA, gene-specific primers were designed based on the gDNA sequence of the gene (Table 2). *CsBAK1* isolated from gDNA was verified and named *gCsBAK1*. The *gCsBAK1* PCR product was likewise cloned into the pCAMBIA1301 vector (named 35S:: gCsBAK1).

**Table 2.** Cloning primer sequences for *BAK1* amplification from genomic DNA.

| Gene | Primer (5′-3′) |
|------|----------------|
| *gBAK1-F* | GGACGAGCTCGGTACCATGTCGGACGATGAAAACGA |
| *gBAK1-R* | GCCCTTGCTCACCATATCACAGTGAGGCAGTATCTGATT |

The CDS sequence of the *BAK1* gene was translated into putative protein sequences. The homology of the gene sequence and cDNA sequence were analyzed by DNAMAN5.0. The transmembrane domain (http://smart.embl.de/, accessed on 26 May 2021) of BAK1 was further analyzed.

### 2.5. Transient Expression of CsBAK1 in'Bingtang' Sweet Orange Leaves

For transient expression, the full length ORF of *CsBAK1* was amplified and cloned into the pCAMBIA1301-sGFP vector, under the control of the 35S promoter. Recombinant plasmids were transformed into the *Agrobacterium* strain EHA105 and then agroinfiltrated into 'Bingtang' sweet orange leaves, as described previously.

Leaves inoculated by injection of *Xcc* ($10^5$ cfu/mL) were randomly sampled for quantification 1 day after infiltration with EHA105. *Xcc* injected leaves (*Xcc* $10^5$ cfu/mL) were co-infiltrated with EHA105 1, 4, and 6 dpi, and leaves were sampled. Sampled leaves were surface disinfected with 75% ethanol, then eight leaf disc samples were collected from within the infiltration areas of each of the three inoculated leaves and homogenized in 1 mL of sterile distilled water. The suspension was then serially diluted with sterile distilled water. 20 μL of the diluted suspension from each sample was incubated on plates of solid LB medium at 28 °C for 48 h. The bacteria were quantified according to the formation of colonies on the LB agar plates. Quantifications were repeated three times.

### 2.6. Construction of BAK1 Promoter Vectors and 'Bingtang' Sweet Orange Leaf Transformation

The *CmBAK1* promoter CmP1 (−2000 bp~+0 bp) was amplified from genomic DNA with primers CmP1-F and CmP1-R. Then, a series of nested deletion fragments—CmP1 (−2000 bp~+0 bp), CmP2 (−1571 bp~−0 bp), CmP3 (−839 bp~−0 bp), CmP4 (−560 bp~−0 bp), and CmP5 (−2000 bp~−1574 bp) were amplified from PCX-CSP1 using ordinary reverse primers −1161 bp (Table 3), and the full-length promoter and 5 '-deleted derivative were cloned into PCX-GUS upstream of β-glucuronidase. Empty PCX-GUS vector was used as a negative control. All constructs were transformed into *Agrobacterium* EHA105 then infiltrated into 'Bingtang' sweet orange leaves according to the previously described protocol. One day after infiltration, *Xcc* ($10^5$ CFU/mL) was randomly selected to inject leaves for quantification.

**Table 3.** Primers used for promoter cloning.

| Gene | Primer (5′-3′) |
| --- | --- |
| *CmP1-F* | AGGATCCCCAATACTTTGTTCAAAGCTGGGTCAAACC |
| *CmP1-R* | TGGATCCCCAATACTGTTCATCCAACTAATCTGATCTTCT |
| *CmP2-F* | AGGATCCCCATACTGACATCATCAATTCATAGTTCAGGT |
| *CmP3-F* | AGGATCCCCAATACTGTAATTTCCAACGTCGCACTTT |
| *CmP4-F* | AGGATCCCCAATACTCGTGAACACTAAATAACAACATTT |
| *CmP-R* | TGGATCCCCAATACTTATTGAAATACAAGTAATGCAGGT |

GUS expression was detected according to the method described by Jefferson et al. [18]. Citrus leaves carrying different promoter CmP fragments were treated with GUS staining solution (50 mM PBS, pH 7.0, 10 mM EDTA, pH 8.0, 20% methanol, 0.1% Triton X-100, 0.1% sodium lauryl sarcosine and 10 mM β-mercaptoethanol). The samples were vacuumized for 1–2 h and incubated at 37 °C for 12–16 h. The dye was decolorized with 70% ethanol for 1–2 day, and then the staining was observed and photographed.

### 2.7. Yeast One-Hybrid (Y1H) Screening

Y1H screening was conducted with the Matchmaker Gold OneHybrid Library Screening System (Takara, Clontech, Cat. Nos. 630491, Dalian, China). A 426 bp fragment of the bait sequence was cloned into the pAbAi vector containing the AUR1-Cgene, conferring resistance to Aureobasidin A (AbA, acyclic depsipeptide used to select yeast). A bait-specific reporter strain was then generated by homologous recombination using the resulting pAbAi-Bait construct (Table 4). Aureobasidin A was found to inhibit the bait-specific reporter strain at a minimal inhibitory concentration. Along with screening a cDNA library generated from Citron C-05 leaves using the strain, transformants grown on selective medium (SD/Leu/AbA 250 ng/mL) were screened and positive colonies were identified by PCR. DNA sequence retransformation assays were carried out by amplifying full-length CDSs from cDNA using the primers listed in Table 5. PCR products were cloned

into the pGADT7 vector, and a bait reporter yeast strain was used to transfer the constructs. After being cultured on SD/Leu and (SD/Leu/AbA 250 ng/mL) plates for 3 day at 30 °C, the cells were resuspended in liquid media to an OD600 of 0.1 ($10^{-1}$) and diluted in a 10-fold dilution series ($10^{-2}$ to $10^{-3}$). Each dilution was spotted seven times on AbA media selective for the respective plasmids (SD/−Leu)) and interactions (SD/−Leu)/AbA 250 ng/mL), complemented with 100 ng/mL AbA to suppress background growth.

**Table 4.** Primer sets used in bait promoter cloning.

| Gene | Primer (5′-3′) |
| --- | --- |
| *BAK1-CmP4-F* | gaaaagcttgaattcgTAAGTATTATTGTATTGTTCAAAGCTGG |
| *BAK1-CmP4-R* | atacagagcacatgccATATTGAAATACAAGTAATTTGCAGGT |

**Table 5.** Primer sets used in *RAP2-13* cloning.

| Gene | Primer (5′-3′) |
| --- | --- |
| *RAP2-13-AD*-F | gtaccagattacgctcaATGGCGGCTACAATGGATTT |
| *RAP2-13-AD*-R | atgcccacccgggtggTTAAGATAATATTGAAGCCCAATCAAT |

### 2.8. Expression Profiling of Transcription Factors

The expression of defence genes (*CmRAP2-13*) induced by *Xcc* was evaluated. Primer sequences for *RAP2-13* was selected from Table 6.

**Table 6.** The primer sequences of *RAP2-13*.

| Gene | Primer (5′-3′) |
| --- | --- |
| *RAP2-13*-F | CCGTTTGGTGGTGAACTTATGG |
| *RAP2-13*-R | GAGGAGGTGGGTAAGACTGGTAAT |

### 2.9. Data Processing

Microsoft Excel 2016 Software (16.0.4266.1001) was used for data processing, GraphPad Prism 7 Statistical Analysis Software was used for variance analysis and significant difference analysis, and the Duncan method was used for multiple comparisons.

## 3. Results

### 3.1. Canker Symptom Development on Leaves of Susceptible and Resistant Sweet Orange Cultivars

Fully expanded young leaves were infiltrated with $10^5$ cfu/mL of *Xcc* on the left of the main vein and with the same concentration of *Xoo* on the right, and the incidence of citrus germplasm was observed regularly (Figure 1). At 6 days after *Xcc* inoculation, the leaf of Citron C-05 began to show tissue depression, which was the disease resistance reaction, while the leaf symptoms of the susceptible genotypes (Nanchuan citron, Round citron, Wild citron, sweet orange, and pomelo) were obviously observed. Leaves of Finger citron and American citron appeared some reaction points. On the 8th and 10th day after *Xcc* inoculation, the tissue sag of Citron C-05 increased, and the symptoms of susceptible cultivars increased were aggravated, and the callus swelling developed, which was the typical symptom of citrus canker disease (Figure 1).

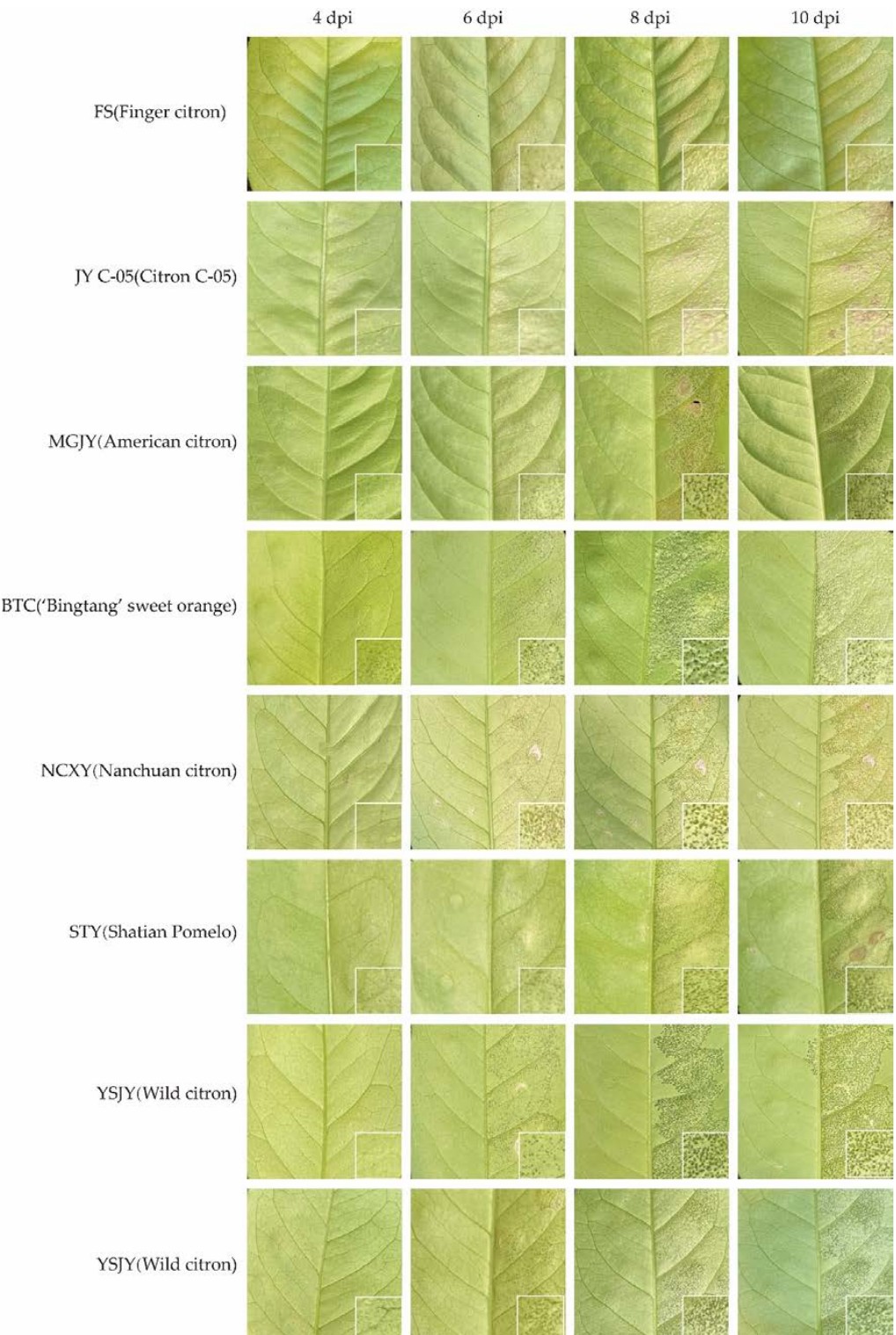

**Figure 1.** Leaf symptom development of tested citrus cultivars following inoculation with *Xcc*. Citron C-05, American Citron, Nanchuan Citron, Round citron, sugar orange, Shatian pomelo, and wild citron. Citron fully expanded young leaves were infiltrated with $10^5$ cfu/mL of Xcc on the left of the main vein and with the same concentration of Xoo on the right. Symptoms were observed at 2, 4, 6, 8, and 10 dpi.

### 3.2. Analysis of CsBAK1 Expression Levels in Citrus Leaves

Quantitative PCR (qPCR) results showed that *BAK1* was up regulated after *Xcc* inoculation in both resistant and susceptible genotypes. However, while *BAK1* was significantly up-regulated in resistant genotypes, while only weakly up-regulated occurred in the susceptible genotypes. Crucially, *BAK1* in the resistant Citron C-05, Finger citron, and American citron began to be highly expressed at 4 dpi and its expression level peaked at 8 dpi. In susceptible citrus cultivars, *BAK1* was not significantly up-regulated until 10 dpi (Figure 2). These results indicated that *BAK1* was specifically upregulated by the citrus canker bacteria in the resistant germplasm, and thus might play a key role in the resistance response to *Xcc*.

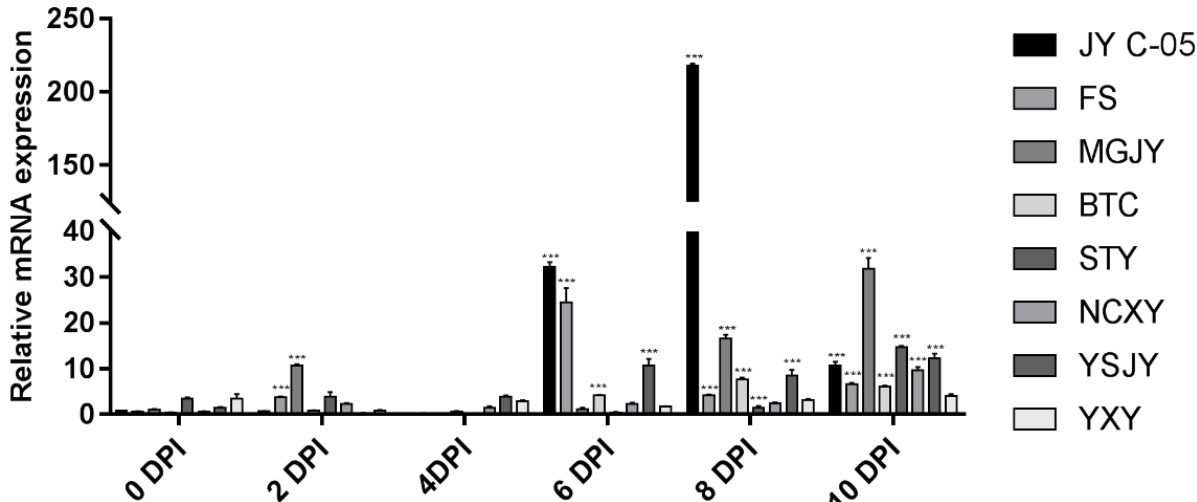

**Figure 2.** Relative expression levels of *BAK1*. *** indicate significant differences ($p < 0.05$, respectively). BTC: 'Bingtang' sweet orange; STY: Pomelo; NCXY: Nanchuan citron; YSJY: Wild citron; YXY: Round citron; MGJY: American citron; FS: Finger citron; JY C-05: Citron C-05.

### 3.3. Isolation and Phylogenetic Analysis of the BAK1 Gene from the Citron C-05 Cultivar

To verify whether there was an association between the BAK1 protein structure and resistance of Citron C-05 to *Xcc*, the BAK1 amino acid sequences were compared across the resistant Citrons (Citron C-05, Common citron, American citron, Aiguo citron) and the susceptible citrus genotypes ('Bingtang' sweet orange, Danna citron, Nanchuan citron, Wild citron, Round citron, and Small citron) obtained by resequencing. The results suggested that BAK1 had 6 amino acid site differences, but there was no obvious correlation between these differences and resistance. The BAK1 protein structure, predicted by SMART online software, inferred that its secondary structure likely consists of two transmembrane regions, six leucine repeats, one low repetition region, and one protein kinase domain (Figure 3). The secondary protein structure of the tested citrus taxa did not reveal significant differences. Phylogenetic analysis showed that CsBAK1 shared high homology with BAK1 (Figure 4).

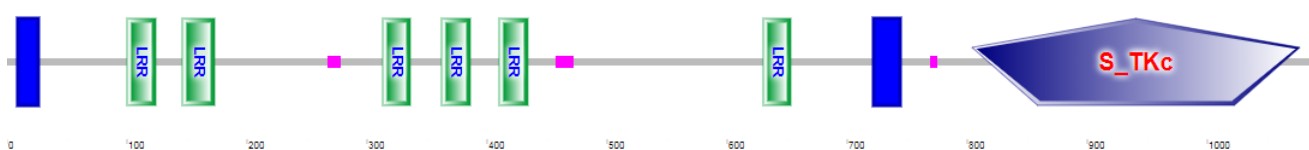

**Figure 3.** Secondary structure of BAK1. LRR: leucine-rich repeats motif; S_TKc: Tyrosine kinase, catalytic domain.

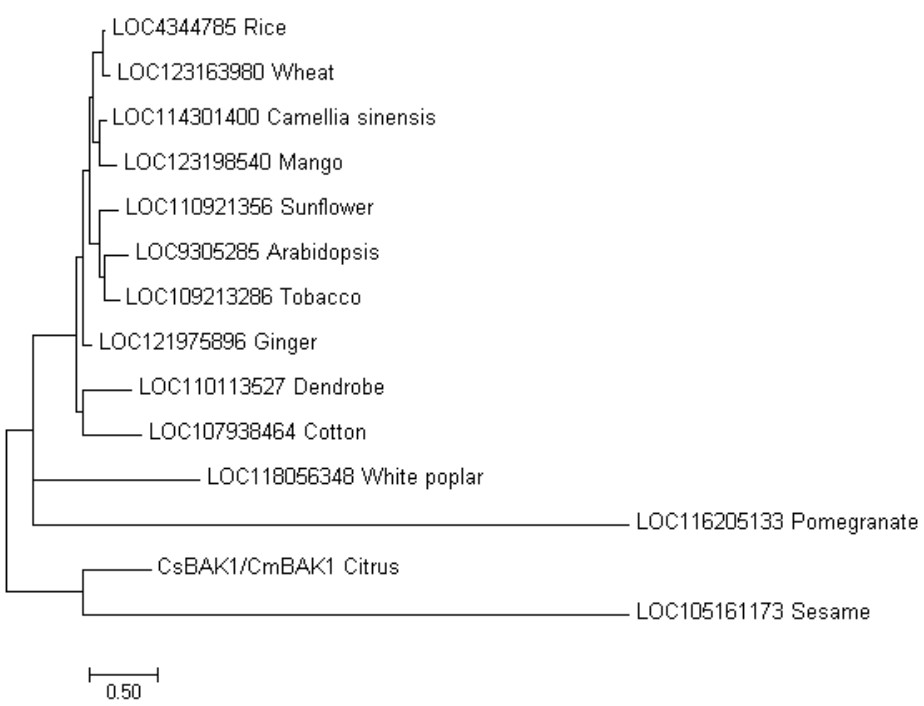

**Figure 4.** Phylogenetic analysis of BAK1 in different citrus species.

*3.4. Transient Expression of CsBAK1 in 'Bingtang' Sweet Orange Leaves*

Disease symptom development was observed in *CsBAK1* leaves every day after inoculation with *Xcc*, and the bacterial load of *Xcc* in inoculated leaves was quantified at 4 and 6 dpi. The most severe symptoms appeared around the injection site, producing a large number of disease spots. When *A. tumefaciens* containing the *BAK1* gene expression vector was infiltrated into inoculated leaves, the symptoms were much reduced, with only sporadic disease spots developing on the leaf surface. The pathogen load in leaves was consistent with the observed symptom severity. The number of canker bacteria per unit leaf area was highly correlated with the severity of symptoms at 4 and 6 dpi. The leaves infiltrated with the *BAK1* expression vector presented mild symptoms and the bacterial content was significantly lower than that of the control, indicating that BAK1 likely induced a bacteriostatic effect (Figures 5 and 6).

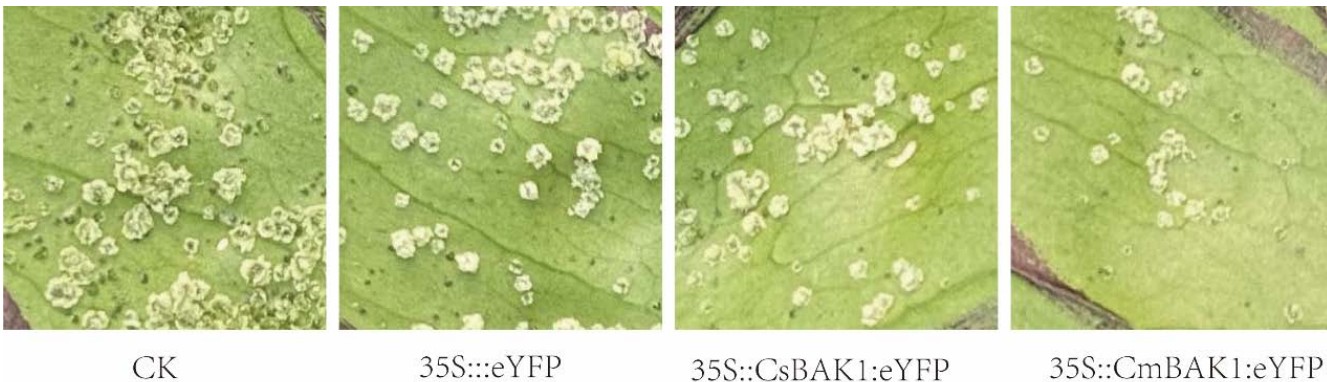

**Figure 5.** Leaf symptoms after transient expression of *BAK1*. **CK:** control injected with buffer solution. **35S::eYFP**: control injected with *Agrobacterium* of empty vector. **35S::CsBAK1::eYFP**: leaves injected with *Agrobacterium* of empty vector 35S:: CsBAK1::eYFP. **35S::CmBAK1**::eYFP leaves injected with *Agrobacterium* of empty vector 35S::CmBAK1::eYFP.

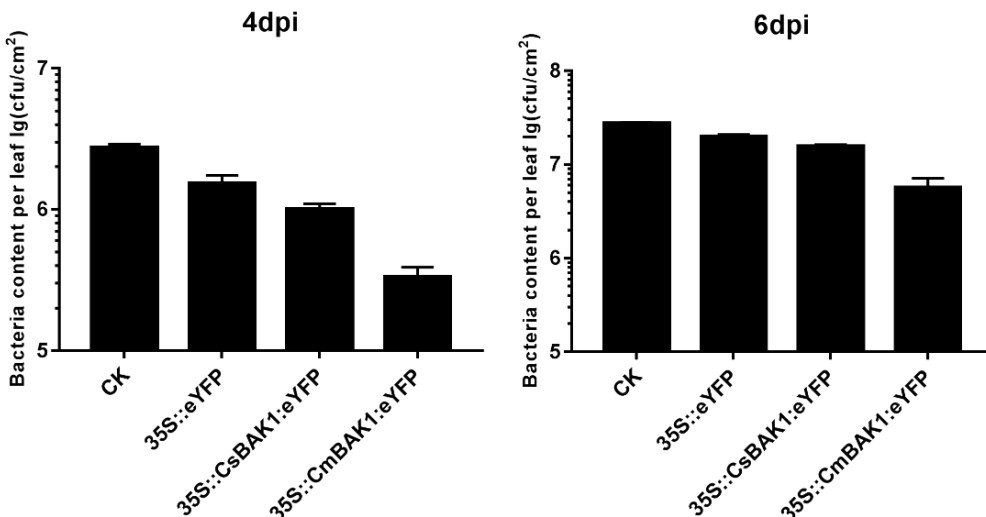

**Figure 6.** *Xcc* growth in each leaf after transient overexpression of *BAK1.* **CK:** control injected with buffer solution. **35S::eYFP**: control injected with *Agrobacterium* of empty vector. **35S::CsBAK1::eYFP**: leaves injected with *Agrobacterium* of empty vector 35S::CsBAK1::eYFP. **35S::CmBAK1**::eYFP leaves injected with *Agrobacterium* of empty vector 35S::CmBAK1::eYFP.

### *3.5. Analysis of the BAK1 Promoter Region*

To analyze the crucial regions of the *BAK1* gene promoter, *BAK1* gene promoter sequences were retrieved from resequencing data. 2000 bp upstream of the ATG start codon was considered to contain the promoter sequence. Comparative analysis showed that the promoter sequences were quite different. Moreover, there were correlative differences between the different promoter sequences and *Xcc* resistance.

Using PLACE (http://www.dna.affrc.go.jp/PLACE/, accessed on 21 October 2021) software, possible cis-acting elements in the *BAK1* gene promoter were predicted. Here, we used the susceptible citrus germplasm ('Bingtang' sweet orange) and wild citron, resistant citrus (American citron and citron C-05). Sweet orange is a commercial variety cultivated in large quantities and is susceptible to disease. American citrate and wild citrate are closely related to citrate C-05, and thus can eliminate the influence of differences in genetic background. To analyze the crucial regions of the BAK1 gene promoter, we identified differences in the cis-acting elements of the promoter. The cis-acting element TCT-motif was different between resistant and susceptible citrus (Table 7).

The *BAK1* gene promoter can be induced by the citrus canker bacteria. In GUS reporter assays, it was observed that when the promoter pCs-BAK1 of 'Bingtang' sweet orange was expressed, no blue spots developed in the leaves of citrus inoculated with *Xcc*, whereas when that of Citron C-05 was expressed, a large number of blue spots appeared on the leaves (Figure 7), suggesting that the promoter pCs-BAK1 of sweet orange could not be activated by *Xcc*, while that of Citron C-05 could. To identify the core region of promoter driven activity, plant expression vectors with truncated 5 ends were designed and constructed according to the distribution of promoter core elements (Figure 7). The constructed vectors were transformed into *Agrobacterium tumefaciens* EHA105. The promoter activity was verified by GUS staining after transient expression in citrus leaves. From test of promoter 5′ deletions, it was found that the core region of *BAK1* promoter activity was located between 2000 bp and 1161 bp.

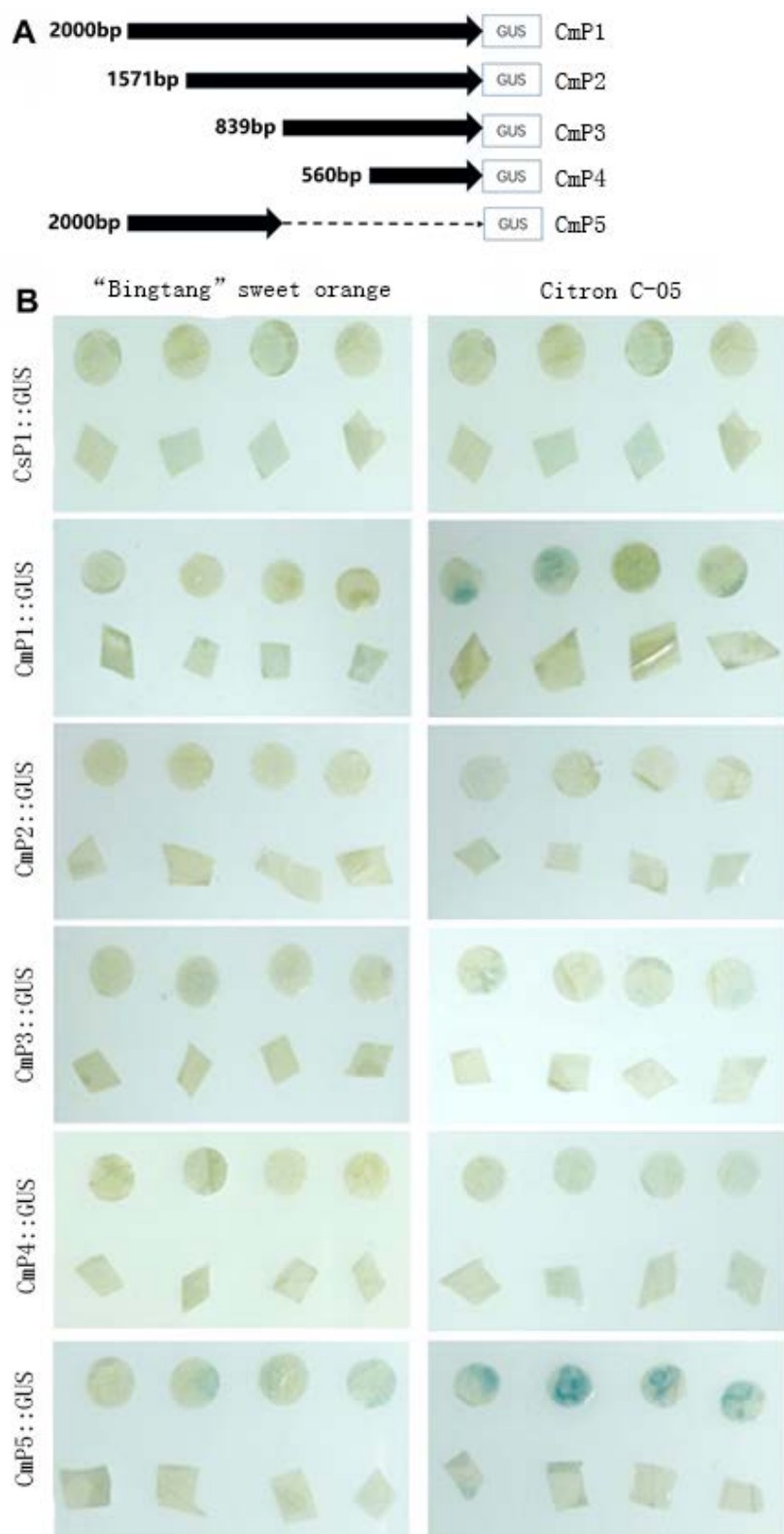

**Figure 7.** GUS expression analysis of transiently expressed truncated BAK1 promoters. (**A**) Schematic representation of the deletion promoter constructs. (**B**) GUS staining assay of sweet orange leaves upon ectopic expression of the promoter: GUS post inoculation with *Xcc*. Circular section: treatment; Triangle: control. CmP1: The full-length CmBAK1 gene promoter (−2000 bp to +0 bp); CmP2: (−1571 bp to −0 bp); CmP3: (−839 bp to −0 bp); CmP4: (−560 bp to −0 bp); CmP5: (−2000 bp to −1161 bp).

**Table 7.** Cis-acting elements and functions of the *BAK1* promoter. BTC: 'Bingtang' sweet orange; YSJY: Wild citron; MGJY: American citron; JY C-05: Citron C-05.

| Cisacting Element | Position (bp) | | | |
|---|---|---|---|---|
| | Susceptible Citrus Germplasm | | Resistant Citrus | |
| | BTC | YSXY | MGXY | JY-C05 |
| CCAAT-box | | | −207 | |
| TCT-motif | | | +1074, +1452 | +636, +1004, +1452 |
| WUN-motif | +868, +1676 | +903, +1672 | −106, +1671 | +883, +1671 |
| MYB | +946 | | +302 | |
| ABRE | +1570 | − | − | − |
| G-box | +784, +1569 | +819, +1105 | −1573 | +799, +1086 |
| Box 4 | +59, +188, +192, −1374, −1707 | +93, +222, +226, −1379, −1703, −1709, −1713 | +68, +270, +511, +640, +644, −1270, −1708, −1712 | +73, +202, +206, −1378, −1708, −1702, −1712 |
| DRE | | −1511 | −1510 | |
| ARE | | +52, +1391, +1068, −1806, −1921 | +470, +1390, +1806, −1921 | +32, +1364, +1390, +1608, −1806, −1921 |
| P-box | +97 | +131 | +549 | +111 |
| W box | −14, −198, +742, −1638 | −48, −232, +777, −1633 | −446, −650, −1195, −1632 | −28, −212, +757, −1632 |
| TCA-element | +709, +769 | | +1162 | |
| AT-rich element | +1862 | | +1862 | +1862 |
| MYC | +797 | +832 | + | +812 |
| ERE | +203, +1210, −1279 | +237, −1226, −1309 | +655 | +217, −1207, −1290 |
| STRE | | +250, +1640 | +668, +1639 | +230, +1639 |
| TC-rich repeats | −35, −589 | −624 | −487, −1042 | −49, −1197 |

### 3.6. Identification of BAK1 Promoter Binding Proteins

To identify proteins that can bind to the BAK1 promoter region, the 426 bp fragment mentioned above was used as a bait to screen transformants from a cDNA library generated from the leaves of plants infected with PGADT for 24 h by the yeast one-hybrid (Y1H) screening system (Figure 8). The monoclonal yeast plasmid of primary screening sequencing was retransformed into baited yeast. Most of the primary screening clones could not produce single colonies on the screening medium, but the plasmid of the growing colony was sequenced. After comparing the sequences in the NCBI database, these genes were named respectively. The transcription factor found to interact with the BAK1 promoter was CmRAP2-13.

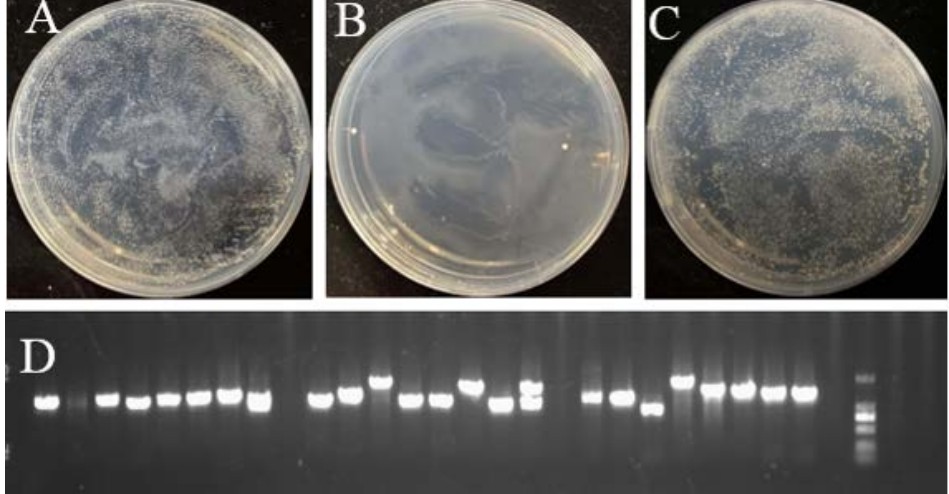

**Figure 8.** Screening of cDNA library of Citron C-05. (**A**) Y1HGold[pBait-AbAi] yeast cells were transformed with a prey vector (CmP5); (**B**) Y1HGold[pBait-AbAi] yeast cells were transformed with a prey vector containing the transcription factor fused to a GAL4 activation domain; (**C**) Preliminary screening and rescreening; (**D**) PCR was used to detect the inserted fragment (containing the transcription factor) in the product sequencing alignment.

All the selected transcription factors and promoters grew well on medium free of basiditin, but differences were found after addition of the selected concentration of basiditin. Among the promoter fragments of *BAK1*, the fragment of Citron C-05 could interact with transcription factor CmRAP2-13 and permit colony growth, whereas the fragment of the 'Bingtang' sweet orange promoter could not (Figure 9).

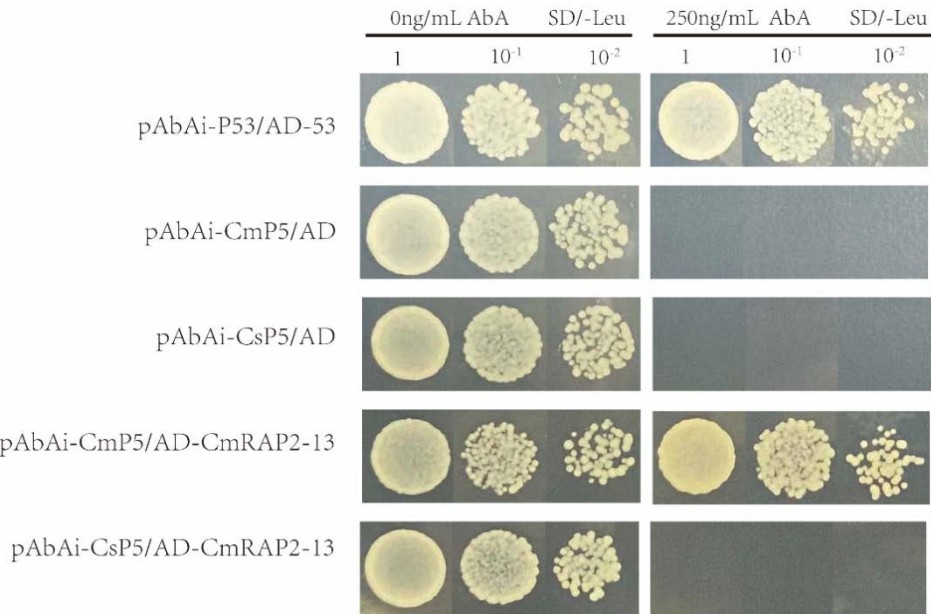

**Figure 9.** Promoter interactions with selected transcription factors. TFs (RAP2-13) bind to the BAK1 promoter in yeast. Bait strain Y1HGold[pBait-AbAi] yeast cells were transformed with a prey vector containing RAP2-13 fused to a GAL4 activation domain. Cells were grown in liquid media to $OD_{600}$ of 1 ($10^{0}$) and diluted in a $10\times$ dilution series ($10^{-1}$ to $10^{-3}$). For each dilution, 10 μL was spotted on media selecting for both plasmids (SD/−Leu) and selecting for the interaction (SD/−Leu/AbA$^{250}$), supplemented with 100 ng mL$^{-1}$ AbA to suppress background growth.

### 3.7. RAP2-13 Gene Expression in Citrus Leaves

Quantitative analysis showed that the expression of *RAP2-13* was significantly up-regulated in resistant germplasm at 6 dpi with *Xcc* and peaked at 10 dpi, whereas it was not significantly up-regulated in susceptible genotypes (Figure 10). The expression of RAP2-13 induced by the citrus canker pathogen indicated that RAP2-13 plays a role in the resistance of Citron C-05 and other resistant germplasm to *Xcc*.

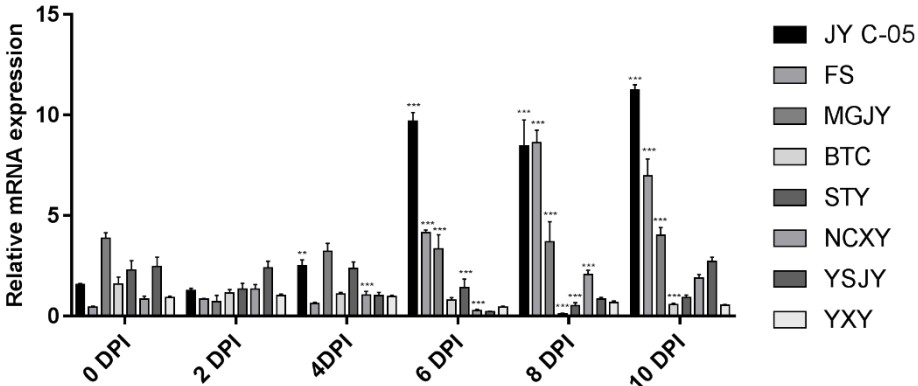

**Figure 10.** Relative expression levels of transcription factor RAP2-13. ** and *** indicated significant differences ($p < 0.05$, respectively). BTC: 'Bingtang' sweet orange; STY: Pomelo; NCXY: Nanchuan citron; YSJY: Wild citron; YXY: Round citron; MGJY: American citron; FS: Finger Citron; JY C-05: Citron C-05.

## 4. Discussion

Plants have developed a sophisticated innate immune response that can quickly sense and defend against pathogens and limit their proliferation [19,20]. Compared to the immune systems of animals, plants lack mobile defence cells and adaptive immune systems. Instead, plants have evolved their innate immune systems to recognize various molecules produced by pathogens or plant cells infected with pathogens, and subsequently activate defensive responses. These microbial molecules are called pathogen-associated molecular patterns PAMPs [21–23]. The receptor kinase BAK1/SERK3 and its approximate BAK1-like protein 1 (BKK1/SERK4) act as receptors for LRR-RLK type pattern recognition responses (PRRs), such as flagellin receptor FLS2 [8] and the bacterial extension factor EF-TU receptor EFR [24]. Biochemical and structural analysis showed that bacterial flagellin epitopes flg22 induced the formation of the FLS2-BAK1 heterocomplex within seconds through direct interaction with LRR domains of FlS2 and BAK1. The C-terminal fragment of flg22 seems to be the molecular glue of the FLS2-BAK1 complex [25]. In this study, *BAK1* in the resistant citrus cultivars Citron C-05, Finger citron, and American citron became highly expressed at 4 dpi with *Xcc*, with expression peaking at 8 dpi. Leaves injected with *BAK1* expression vector harboring *Agrobacterium* presented mild symptoms and the bacterial content of inoculated *Xcc* was significantly lower than that of the control, indicating that BAK1 exerted a bacteriostatic effect. Sequence analysis and protein structural analysis showed that both BAK1 proteins contained multiple LRR structures, which may be involved in the PTI response. Formation of this core receptor recognized by flg22 leads to trans-phosphorylation of both proteins, which is critical for mediating downstream signaling [25,26]. For BAK1 and CERK1 coreceptors, their kinase activity is essential for mediating downstream signals [27,28].

The induced plant defence response is the result of the combined action of inducer promoters, various relevant cis-regulatory elements, signal transduction pathways, and pathogen-specific responses [29]. Gene promoters induced by pathogen elicitors or pathogen attack are heretofore referred to as "pathogen-sensitive promoters" or "pathogen-induced promoters" [30]. The regulatory mechanisms of these promoters also differ depending on the presence of pathogens and specific regulatory elements. Several genes and their promoters play crucial roles in jasmonic acid-mediated defence signaling pathways against pathogen attack. The findings of Zhou et al. [31] showed that *Xoo* stimulated the inhibition of JA biosynthesis through "SAPK10-WRKY72-AOS1" module infection, leading to increased susceptibility to *Xoo*. The key to promoter regulation lies in the transcriptional levels of downstream regulatory gene expression, and the role of promoters in the process of acquiring resistance in plants infected with pathogenic microorganisms has been studied. In this study, we cloned the *BAK1* gene promoter of the susceptible 'Bingtang' and resistance Citron C-05 sweet orange cultivars, analyzed the sequence of the promoter in the resistant germplasm, and analyzed the cis-acting elements of the promoter, including WRE3, CCAAT-box, MYB recognition site, G-box, W box and ABRE. The cis-acting element TCT-motif was different between resistant and susceptible citrus. However, whether the differences in these components cause the differences in promoter activity needs to be further investigated.

In the present study, CmRAP2-13 was identified as a transcriptional activator of *CmBAK1*. CmRAP2-13 is a member of the AP2/ERF transcription factor family, which directly interacts with cis-acting elements such as GCC-box on target gene promoters [32]. Studies have shown that ERF2 promoter-binding transferrin transcription factors can positively regulate the production of plant disease resistance proteins in *Streptomyces attenuatum*, and this protein plays an important role in *Streptomyces* infection [33]. The AP2/ERF superfamily transcription factors regulate many processes in plant development and play an important role in hormone regulation and stress responses. The spatiotemporal expression of the CmRAP2-13 transcription factor regulated by the citrus canker pathogen exhibited differences between 'Bingtang' and Citron C-05 sweet orange cultivars, and our promoter analysis also found that there were different loci.

## 5. Conclusions

In this study, we identified the transcription factor CmRAP2-13 plays an important role in resistance to the citrus canker pathogen *Xanthomonas citri* subspecies *citri* (*Xcc*). In response to *Xcc* invasion, CmRAP2-13 stimulates a resistance response by directly binding to the *CmBAK1* promoter to activate *CmBAK1* expression. Our findings help to elucidate the regulatory mechanism of *BAK1* gene in *Xcc* resistance and expand the current understanding of the function of *BAK1* in plant pathogen defences.

**Author Contributions:** Conceptualization, Q.W., M.Z., Y.L., D.L., X.M. and Z.D.; Data curation, Q.W., M.Z., Y.L., D.L., X.M. and Z.D.; Formal analysis, Q.W., M.Z. and Y.L.; Investigation, Q.W., M.Z. and Y.L.; Methodology, Q.W., M.Z. and Y.L.; Resources, Q.W., M.Z. and Y.L.; Software, Q.W.; Supervision, Q.W. and M.Z.; Visualization, Q.W., M.Z. and Y.L.; Writing—original draft, Q.W., M.Z. and Y.L.; Writing—review & editing, Q.W., M.Z., Y.L., D.L., X.M. and Z.D. All authors have read and agreed to the published version of the manuscript.

**Funding:** This work was supported by the Key Project of International Cooperation and Ex-change of the National Natural Science Foundation of China (No. 31720103915).

**Data Availability Statement:** Data are contained in the article.

**Conflicts of Interest:** The authors declare no conflict of interest.

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
