# Peer review of "Identification of the Transcription Factors RAP2-13 Activating the Expression of CsBAK1 in Citrus Defence Response to Xanthomonas citri subsp. citri"

_horticulturae, doi:10.3390/horticulturae8111012_

Round 1

Reviewer 1 Report (Previous Reviewer 1)

The manuscript entitled “Identification of the transcription factors RAP2-13 activating the expression of CsBAK1 in citrus defence response to Xathomonas citri subsp. Citri” is interesting; However, it is necessary improve some issues:

 L82-90. Explicitly write the purpose of the manuscritpt

L101. “previously” is repeated

L203-206. This is methodology

L256. A dot is missing

L361. Use the correct format to cite

Author Response

Reviewer 2 Report (New Reviewer)

The overall quality of the manuscript is good. Only small mistakes are found and can be corrected accordingly. 

Author Response

Reviewer 3 Report (New Reviewer)

The article “Identification of the transcription factors RAP2-13 activating the expression of CsBAK1 in citrus defence response to Xathomonas citri subsp. citri ” is devoted to the important and acute theme of the role of plant transcription factors in plant resistance to these dangerous pathogen. The authors provided a lot of information on molecular participants of this system, but there are some comments:

Title: Xanthomonas!

1)Introduction should be built from General to specific (paragraphs about Xanthomonas, citruses and immunity before section about BAK1, lines 69-75 after line 52). Introduction cannot include conclusions and results (lines 82-90). Authors should give the aim of the investigation.

2)Results must be seriously improved. All codenames must be uniformed on all figures. All figures must be accurately described in the text and accompanied with links in the text.

Fig 1: “… symptoms also developed in the Finger citron and American citron cultivars, however the Citron C-05 cultivar produced a hypersensitive response.” Authors cannot say about hypersensitive responses without some evidence (HSR associated genes transcription, ROS generation). Illustrations of symptoms on 0-4 dpi are almost the same. And I cannot understand how Citron-C05 differs from other ones? Finger citron, for example, do not have some symptomes on this figure.

Table 4: Description “Using PLACE (http://www.dna.affrc.go.jp/PLACE/) software possible cis acting elements in the BAK1 gene promoter were predicted. There were g-box, W box, WUN Motif and other cis-acting elements in the promoter sequences of the BAK1 gene in both resistant and susceptible germplasms, and W-box, WUN, and WRE3 motifs were correlated with disease resistance” is unclear. What type of correlation is observed? What resistant and susceptible germplasms are used and why? 

Fig. 7: What are A and B?

Fig 8: “Figure 8. Screening of cDNA library of Citron C-05: A: Yeast transformation; B、C: Preliminary screening and rescreening;D:PCR was used to detect the inserted fragment” is incorrect legend. Provide lettering description about microbes on plates and bands on PAAG.

Fig 9: Provide description in the text. “Figure 9. Promoter interactions with selected transcription factors.The full-length CmBAK1 gene promoter named CmP1(−2000 bp to +0 bp). CmP5(−2000 bp to -1161 bp)” is incorrect legend, I see microbes on plates, not molecular interaction. 

Fig 10:  Figure 10. “Transcriptome data analysis of promoter interactions with transcription factor RAP2-13.* and ** indicated significant differences (p<0.05 and p<0.01, respectively).” is an incorrect legend. I see only transcription of RAP-13.

Round 2

Reviewer 3 Report (New Reviewer)

Dear authors, 

1)I recommend deletion of symptoms on 0-4 dpi and increase the size of 4-6 dpi simptoms and insert abbreviations (for example, finger citron (FS)).

2)You should place your ansver in the text to eplain Table 4 (To analyze the crucial regions of the BAK1 gene promoter, we identified differences in the cis-acting elements of the promoter. The cis-acting element TCT-motif was different between resistant and susceptible citrus. However, whether the differences in these components cause the differences in promoter activity needs to be further investigated. Here, we used the susceptible citrus germplasm (‘Bingtang’ sweet orange) and wild citron, resistant citrus (American citron and citron C-05). Sweet orange is a commercial variety cultivated in large quantities and is susceptible to disease. American citrate and wild citrate are closely related to citrate C-05, and thus can eliminate the influence of differences in genetic background). I think the sentence “W-box, WUN, and WRE3 motifs were correlated with disease resistance”should be revritten (discussion on correlations demands statistical analisys), and resistant\susceptible varieties should be marked in the table. 

Author Response

This manuscript is a resubmission of an earlier submission. The following is a list of the peer review reports and author responses from that submission.

Round 1

Reviewer 1 Report

 The manuscript entitled “Identification of one transcription factors RAP2-13 activating the expresión of CsBAK1 in citrus defense response Xanthomonas citri subsp. citri" is interesting. However, the document has serious deficiencies that must be addressed before publishing the manuscript, which are shown below:

L10-78. The objective is missing.

L32. Remove comma after “disease is”

L77. CFU/mL or spores/mL? How did you determine the cfu/mL?

L96-157. What were the sequences of the primers used?

L99. E. coli in italics.

L166-169. This is methodology.

Figures 2 and 8 should be improved.

L333. Define Xoo

L349. The conclusion is missing. Because there are no objectives, it is very difficult to complete the work.

L350-355. The funding information, authors’ contributions, conflict of interest, and ethics statements are missing.

Author Response

Dear Editor and Reviewers,

Thanks very much for taking your time to review this manuscript. I really appreciate all your comments and suggestions! Our responses to the reviewers’ comments are detailed below, and changes are highlighted in the revised manuscript. 

COMMENTS TO THE AUTHOR:

Reviewer #1:

Comment1:

L10-78. The objective is missing.

Response1:

The objective: The results of gene function verification by transient expression showed that both CsBAK1 (C. sinensis BAK1)and CmBAK1(C. medica BAK1) could inhibit the growth of the canker pathogen in susceptible genotype ‘Bingtang’ sweet orange to some extent.

Comment2:

L32. Remove comma after “disease is”

Response2:

Thank you for your careful reviewing. We have to modify the article in accordance with the requirements of the reviewers

Comment3:

L77. CFU/mL or spores/mL? How did you determine the cfu/mL?

Response3:

CFU/mL. The OD600 of the bacterial suspension was adjusted to 0.75 (about 109 CFU /mL). After 10-fold gradient dilution, 10-7, 10-8 and 10-9 dilution gradients were selected for plate counting, and 10-4 (2.01×105 CFU /mL) dilution gradient was selected inoculum (see 2.2).

Comment4:

L96-157. What were the sequences of the primers used?

Response4:

We have added primer sequences in the new revised manuscript (Table 1 to 3).

Comment5:

L99. E. coli in italics.

Response5:

Done, thanks.

Comment6:

L166-169. This is methodology.

Response6:

We need to emphasize this point.

Comment7:

Figures 2 and 8 should be improved.

Response7:

We have modified the article in accordance with the requirements of the reviewers.

Comment8:

Define Xoo

Response8:

Xoo (Xanthomonas oryzae pv.oryzae) and Xcc (Xathomonas citri subsp. citri)  belong to Xanthomonas  .Xcc can cause citrus canker .But Xoo can not cause devastating disease for citrus.

Comment9:

The conclusion is missing. Because there are no objectives, it is very difficult to complete the work.

Response9:

The conclusion is that BAK1 proteins contained multiple LRR structures, which may be involved in the PTI response of plant disease resistance.The transcription factors RAP2-13 activating the expression of CsBAK1 in citrus defence response to Xathomonas citri subsp. Citri

Comment10:

The funding information, authors’ contributions, conflict of interest, and ethics statements are missing.

Response10:

Done. Please see funding information, authors’ contributions.

We would like to take this opportunity to offer our gratitude to all of the reviewers for the helpful input about how to improve the scientific rigour and utility of our study and manuscript. Many thanks.

Reviewer 2 Report

The manuscript reports interesting findings on the identification of a transcription factor that interacts with BAK1 in citrus in response to Xcc. However, the data is not well presented in the manuscript. 

The abstract is not clearly written, and certain facts are confusing. Others are not sufficiently described. For example, what is CmBAK1 refer to? What do the authors mean by 'to some extent? 

The authors only briefly introduce the disease in the introduction section. It lacks information on molecular studies on citrus canker. The authors should also include the information available in the literature on the interaction involving BAK1 and other known protein or transcription factors.

The write-up for the material and method section should also be improved. The titles for the method section should be more informative and reflects the content. For example, the title for 2.1 must also mention fungal material. The description of the method is not sufficient. More details should be added for clarity. For instance, from which citrus variety was the BAK1 gene isolated? Which inoculated method was used? There are three different terms to describe the process: agro-infiltration, spray-inoculation, and injection. Please standardize the term used. More information should be added to section 2.7. The name of the citrus varieties mentioned in the method section is different from the varieties listed in the result section. Please identify the susceptible and resistant/tolerant varieties used in the method section. 

Please include a brief description of the objective for each result section. Xoo was mentioned in the result section 3.1 but not in the method and described why it was used as a control. Section 3.3 only describes the findings of protein sequence analysis. Please include the results of gene isolation and phylogenetic analysis. After all, phylogenetic analysis was spelled out in the title. 

Line 230: Instead of directing readers to check the appendix, I recommend the authors elaborate on the difference between the promoter sequences.

Line 261: Is it herbivores-responsive or pathogen-responsive elements?   

The second paragraph of the discussion section is a bit redundant with the information presented in the introduction section. I suggest limiting the background information and focusing on the results obtained by the authors in the discussion section.

Line 338: I don't think this is clearly described in the method section. Please add the information since only Bintang orange was mentioned in the method section. 

Line 345: Only temporal expression was analyzed. Please re-phrase.

In general, more discussion is required to explain the role of RAP2-13. 

There is no conclusion presented, so I cannot comment on that. 

More (recent) references need to be added. There are references that are not properly cited in the text (example, Line 110). 

The authors are required to check the formatting again according to the journal's recommendation. I highly recommend the authors send the manuscript for proofreading since there are plenty of grammatical and sentence structure errors.

Author Response

Dear Editor and Reviewers,

Thanks very much for taking your time to review this manuscript. I really appreciate all your comments and suggestions! Our responses to the reviewers’ comments are detailed below, and changes are highlighted in the revised manuscript. 

COMMENTS TO THE AUTHOR:

Reviewer #2:

Comment1:

The abstract is not clearly written, and certain facts are confusing. Others are not sufficiently described. For example, what is CmBAK1 refer to? What do the authors mean by 'to some extent?

Response1:

I have to modify the article in accordance with the requirements of the reviewers (see the abstract).

Comment2:

The authors only briefly introduce the disease in the introduction section. It lacks information on molecular studies on citrus canker. The authors should also include the information available in the literature on the interaction involving BAK1 and other known protein or transcription factors.

Response2:

Done.Please see see the Introduction.

Comment3:

The write-up for the material and method section should also be improved. The titles for the method section should be more informative and reflects the content. For example, the title for 2.1 must also mention fungal material. The description of the method is not sufficient. More details should be added for clarity. For instance, from which citrus variety was the BAK1 gene isolated? Which inoculated method was used? There are three different terms to describe the process: agro-infiltration, spray-inoculation, and injection. Please standardize the term used. More information should be added to section 2.7. The name of the citrus varieties mentioned in the method section is different from the varieties listed in the result section. Please identify the susceptible and resistant/tolerant varieties used in the method section. 

Response3:

We have revised material and method accordingly. Both susceptible citrus genotypes (‘Bingtang’ sweet orange, Pomelo, Nanchuan citron, Wild citron and Round citron) and resistant ones (American citron, Finger citron and Citron C-05) were chosen in the present study. Xoo (Xanthomonas oryzae pv.oryzae) and Xcc (Xathomonas citri subsp. citri)  belong to Xanthomonas  .Xcc can cause citrus canker .But Xoo can not cause devastating disease for citrus. 10-4 (2.01×105 CFU /mL) dilution gradient was selected inoculum. At the same time, the control was inoculated with the same concentration of Xoo (Xanthomonas oryzae pv.oryzae). CsBAK1 (C. sinensis BAK1)and CmBAK1(C. medica BAK1)

Comment4:

Please include a brief description of the objective for each result section. Xoo was mentioned in the result section 3.1 but not in the method and described why it was used as a control. Section 3.3 only describes the findings of protein sequence analysis. Please include the results of gene isolation and phylogenetic analysis. After all, phylogenetic analysis was spelled out in the title. 

Response4:

Xoo (Xanthomonas oryzae pv.oryzae) and Xcc (Xathomonas citri subsp. citri)  belong to Xanthomonas  .Xcc can cause citrus canker .But Xoo can not cause devastating disease for citrus. We have added the phylogenetic.Comment5:

Line 230: Instead of directing readers to check the appendix, I recommend the authors elaborate on the difference between the promoter sequences.

Response5:

We thank the reviewer for the comments on how to improve our study.We can find difference between the promoter .(Table 4)

Comment6:

Line 261: Is it herbivores-responsive or pathogen-responsive elements? 

Response6:

pathogen-responsive elements.

Comment7:

Line 338: I don't think this is clearly described in the method section. Please add the information since only Bintang orange was mentioned in the method section.

Response7:

I Thank you for your careful reviewing.

Comment8:

Line 345: Only temporal expression was analyzed. Please re-phrase.

Response8:

The interaction between RAP2-13 and Xcc needs further study.

Comment9:

In general, more discussion is required to explain the role of RAP2-13.

Response9:

Thank you for your careful reviewing.Done. Please see discussion.

.Comment10:

There is no conclusion presented, so I cannot comment on that. 

Response10:

The conclusion is that BAK1 proteins contained multiple LRR structures, which may be involved in the PTI response of plant disease resistance.The transcription factors RAP2-13 activating the expression of CsBAK1 in citrus defence response to Xathomonas citri subsp. Citri

Comment11:

More (recent) references need to be added. There are references that are not properly cited in the text (example, Line 110). 

Response11:

Done .

Comment12:

The authors are required to check the formatting again according to the journal's recommendation. I highly recommend the authors send the manuscript for proofreading since there are plenty of grammatical and sentence structure errors.

Response12:

We again thank the reviewer for the comments on how to improve our study. We note that the language has been initially improved by a language service, and now we have sent it for modification by Several researchers, and this language can also be improved by the production editor if it was suitable for publication.We would like to take this opportunity to offer our gratitude to all of the reviewers for the helpful input about how to improve the scientific rigour and utility of our study and manuscript. Many thanks.

Round 2

Reviewer 1 Report

The authors have adjusted the manuscript to make it more understandable. However, some questions remain unanswered, and there are severe deficiencies in the document, such as those shown below:

The paper is not in the format indicated by the authors' guide.

The way of citing other articles in the text is not correct.

It is necessary to review the format of the references section

The objective must be explicitly placed in the document, both in the abstract and at the end of the introduction.

The O.D. indeed has equivalences with the concentration of microorganisms. However, this equivalence concerns cells per mL and not CFUs per mL; this is because the O.D. evaluates all the cells, regardless of whether they are alive or dead. In the case of CFU, it counts the "colony forming units", that is, the cells capable of reproducing (living cells).

L89. Incomplete information was left, with two question marks left in writing.

According to the authors' guide, the conclusion must go in a separate section and must be placed explicitly, not only as a response to the reviewers. The conclusion is an essential part of the document, and the fact that it is not appropriately presented reduces its impact.

Reviewer 2 Report

I thank the authors for the effort to improve the manuscript. But I noticed that the authors did not respond to all the comments. So I am still confused by the information presented in the abstract (and the rest of the manuscript). 

For example, how is C. sinensis BAK1 isolated as a disease resistance-related gene in an interaction between C. medica L and Xcc? Shouldn't it be CmBAK1 isolated from C. medica @ Citron C-05? After all, C. sinensis was never mentioned in the main text. To add to the confusion, the authors mentioned that they cloned BAK1 from Bingtang and Citron C-05 in the discussion section. Hence, my previous question about which citrus variety the BAK1 gene was isolated is still unanswered. 

In addition, despite the statement by the authors that a language service has initially improved the English language in the manuscript and that they have sent it for modification by several researchers; I still find errors in the abstract, not to mention in the main text. 

For example, Line 19 to 25 are pretty hard to follow. 

Line 20: was finally identified (past tense)

Other errors that are still present and easily noticed by glancing through the manuscript: 

Line 89 is missing an information 

Line 110 ampyfied 

Line 188 PCR results 'seemed that' 

Line 248 how can a promoter show blue spots anyway? 

Therefore, in my professional opinion, the manuscript, in its present form, is not suitable for publication in this high-impact journal.